# Effects of Dietary Fatty Acid Composition on Lipid Metabolism and Body Fat Accumulation in Ovariectomized Rats

**DOI:** 10.3390/nu13062022

**Published:** 2021-06-11

**Authors:** Jhih-Han Yeh, Yu-Tang Tung, Yu-Sheng Yeh, Yi-Wen Chien

**Affiliations:** 1School of Nutrition and Health Sciences, Taipei Medical University, Taipei 11031, Taiwan; love21136@gmail.com; 2Graduate Institute of Biotechnology, National Chung Hsing University, Taichung 40227, Taiwan; peggytung@nchu.edu.tw; 3Cell Physiology and Molecular Image Research Center, Wan Fang Hospital, Taipei Medical University, Taipei 11696, Taiwan; 4Graduate Institute of Metabolism and Obesity Sciences, Taipei Medical University, Taipei 11031, Taiwan; 5Department of Medicine, Cardiovascular Division, Washington University School of Medicine, St. Louis, MO 63112, USA; yu-sheng@wustl.edu; 6Nutrition Research Center, Taipei Medical University Hospital, Taipei 11031, Taiwan; 7Research Center of Geriatric Nutrition, College of Nutrition, Taipei Medical University, Taipei 11031, Taiwan

**Keywords:** ovariectomy, monounsaturated fatty acids, body fat accumulation, lipid metabolism

## Abstract

Background: Obesity is a state of excess energy storage resulting in body fat accumulation, and postmenopausal obesity is a rising issue. In this study using ovariectomized (OVX) rats, we mimicked low estrogen levels in a postmenopausal state in order to investigate the effects of different amounts and types of dietary fatty acids on body fat accumulation and body lipid metabolism. Methods: At 9 weeks of age, rats (*n* = 40) were given an ovariectomy, eight of which were sham-operated to serve as a control group (S). We then divided OVX rats into four different intervention groups: diet with 5% soybean oil (C), and diet with 5% (L), 15% (M), and 20% (H) (*w*/*w*) experimental oil, containing 60% monounsaturated fatty acids (MUFAs) and with a polyunsaturated/saturated fatty acid (P/S) ratio of 5. Results: After OVX, compared to the S group, the C group showed significantly higher body weight, and insulin and leptin levels. Compared to the C group, the H group had lower hepatic triglyceride level and FAS enzyme activity, and higher hepatic ACO and CPT-1 gene expressions and enzyme activities. Conclusions: An OVX leads to severe weight gain and lipid metabolism abnormalities, while according to previous studies, high fat diet may worsen the situation. However, during our experiment, we discovered that the experimental oil mixture with 60% MUFAs and P/S = 5 may ameliorate these imbalances.

## 1. Introduction

Obesity is a worldwide health issue, and awareness of it has greatly increased in recent years. When obesity occurs, comorbidities follow, including hyperlipidemia, hyperglycemia, hypertension, and other metabolic abnormalities. White adipose tissues (WATs) are composed of white adipocytes, and play an important role in the body’s energy balance. Adipocytes were found to be over-expanded in obese individuals, while the expansion occurs in two different ways [1]. In principle, hypertrophy is the enlargement of adipocyte size, and hyperplasia is an increase in adipocyte numbers. With the size expansion, the function of adipocytes is gradually lost, which is believed to cause metabolic abnormalities in individuals [2]. For example, dyslipidemia, an abnormal accumulation of one or more kinds of lipoproteins, leads to changes in blood lipid concentrations. Evidence suggests that obesity is caused by excessive energy intake that leads to abdominal accumulation and possibly contributing to peripheral insulin resistance (IR) [3,4].

Torng et al. [5] found that menopause in Taiwanese women had an adverse effect on blood lipids, especially increases in total cholesterol (TC) and low-density lipoprotein cholesterol (LDL-C), and a decrease in high-density lipoprotein cholesterol (HDL-C). In the process of menopause, the concentration of HDL-C changes in a U-shaped curve [5]. The concentration of HDL-C reaches the highest during menopause and begins to decrease after menopause. At the same time, compared to before menopause, LDL-C, which causes atherosclerosis, appears in large amounts in postmenopausal women, resulting in a three-fold increase in the risk of cardiovascular diseases and coronary artery calcification. Based on past human and animal experiments, estrogen is believed to reduce food intake and increase energy consumption in the body, and is therefore related to preventing obesity [6,7]. The reduction in estrogen after menopause affects the location of fat storage in the body, resulting in increases in the overall fat mass and abdominal fat [8]. Under the menopausal animal model, the concentration of free fatty acids (FAs, FFAs) in the body was significantly reduced, but after 60 days of estrogen supplementation, their concentration rose. At the same time, that study further pointed out that the accumulation of abdominal fat may be the result of abnormal lipolysis caused by low estrogen level [9]. An ovariectomy (OVX), surgery to remove the ovaries, is a common menopausal animal model. Removal of the ovaries simulates a state of low estrogen and is often used to observe related effects of metabolism in the body in this state. In previous studies, it was found that 1 week after an OVX, OVX rats had a significant increase in body weight compared to rats that had undergone sham surgery. Even with the same food intake, OVX animals still tended to gain more body weight. This was accompanied by a decrease in energy consumption [10] and accumulation of additional abdominal white fat [11]. In the OVX animal model, low estrogen can easily cause a large amount of fat to accumulate in the liver. In previous literature, the occurrence of this phenomenon was associated with increases in fat synthesis pathways [12] and decreases in liver fat oxidation pathways [13]. At the same time, the loss of the protective effect of estrogen can easily cause an imbalance in the oxidative and antioxidant functions in the liver of OVX animals, resulting in increases in reactive oxygen species (ROS) [14].

There are not many previous studies on the relationship between OVX animal models and the FA composition. Kim et al. [15] reported that n-3 polyunsaturated FAs (PUFAs) given to OVX rats could reduce triglycerides (TGs). In that study, 48 mice were subjected to an OVX, and then after a 1-week recovery period, they were fed an AIN-93G diet the formula of which was adjusted to provide 0%, 1%, or 2% kcal eicosapentaenoic acid (EPA) + docosahexaenoic acid (DHA), in an experiment that lasted 12 weeks. Results showed that the expressions of sterol regulatory element-binding protein (SREBP)-1, acetyl-CoA carboxylase (ACC), FA synthase (FAS), and diacylglycerol acyltransferase 2 (DGAT2) in the liver were significantly reduced. Expressions of AMP-activated protein kinase (AMPK), phosphorylated AMPK, peroxisomal proliferator-activated receptor α (PPARα), and carnitine palmitoyltransferase (CPT)-1 in the liver and skeletal muscles significantly increased. The authors stated that n-3 PUFAs administered to OVX rats reduced TGs, which may have been due to reductions in TG biosynthesis pathways in the liver and increased oxidation of TGs [15]. Liao et al. [16] fed hamsters with different proportions of an FA diet, and found that there were no differences in food intake among the various groups. Hamsters fed experimental oil with a polyunsaturated/saturated FA (P/S) ratio of 5 and a monounsaturated FA (MUFA) ratio of 60% had a lower body weight (BW) and WAT weight, and had lower plasma insulin and free FA concentrations along with liver lipid-synthesizing enzyme activity. Activities of the liver FA oxidation-related enzymes, acyl-CoA oxidase (ACO) and CPT-1, increased, which inhibited body fat accumulation. In 2017, it was further discovered that experimental oil with an FA composition of P/S = 5 and a MUFA ratio of 60% could still effectively reduce lipid synthesis and promote lipid oxidation under a high-dose high-fat diet (HFD) to reduce body fat accumulation. This experimental oil seemed to prevent an increase in BW caused by the intake of medium-fat or HFDs by preventing increases in the insulin concentration, FAS and lipoprotein lipase (LPL) enzyme activities, and peroxisome proliferator-activated receptor (PPAR)-γ and LPL gene expressions [17].

Although a lot of research has been conducted to study the effect of fat and its fatty acid composition on normal animal body weight, fat and glucose metabolism, the effect of fatty acid composition on OVX rats has not been studied. In this study, OVX rats were fed diets containing different doses of a specific proportion of experimental oil to explore its effects on WAT quality, blood lipid concentrations, liver lipid accumulation, and FA metabolism-related enzyme activities and gene expressions. Therefore, the results of this study can provide a reference for how to choose fatty acid composition during menopause.

## 2. Materials and Methods

### 2.1. FA Composition Analysis

Experimental oil was mixed following Liao’s method [18]. Soybean oil and experimental oil mixture were purchased from a local supermarket and analyzed by gas chromatography (GC). Samples were extracted using a modified Folch method [19]. Samples was extracted using chloroform/methanol = 2/1 (*v*/*v*) for 1 h, and then distilled water was added to separate the liquid. The extract was incubated for 10 min at room temperature and centrifuged at 4 °C and 3000 rpm for 10 min. The lower phase was collected. FA methylation was performed by heating the sample to 88 °C with 14% boron trifluoride/methanol (B1252, Sigma, St. Louis, MO, USA) for 1 h to form FA methyl ether (FAME) and then the solvent was removed. FAME was analyzed using a FOCUS™ GC (Thermo Fisher Scientific, Milan, Italy) equipped with a 30-m × 0.32-mm inner diameter (I.D.) × 0.20-μm df Rtx-2330 column (Restek, Bellefonte, PA, USA) and flame ionization detector. 

FA compositions of the soybean oil and experimental oil were analyzed by GC. Results were obtained based on the retention time of the appropriate standard (GLC-455; Supelco, St. Louis, MO, USA), and percentages of the FA profile were calculated based on 16 different FAs, as shown in Table 1.

### 2.2. Animals and Experimental Design

A total of 40 female Sprague-Darley rats were purchased from BioLasco (Taipei, Taiwan) at 9 weeks of age (ethics number LAC-2017-0443, Taipei, Taiwan). The study was conducted under strict guidelines of the institutional animal care and used committee of Taipei Medical University. Thirty-two rats underwent an OVX, while eight were sham-operated, and serum estradiol level was tested to ensure the effects of the surgery. Animals were housed in an air-conditioned room (22 ± 2 °C and 65% ± 5% relative humidity) with a 12 h light/12 h dark circle. All mice have free access to f water and a basic diet (Rodent Laboratory Chow 5001; PMI Nutrition International, St Louis, MO, USA). Rats were then communally housed in plastic cages (three per cage).

After a 4-week surgical recovery period, OVX rats were randomly divided into four different dietary groups (*n* = 8 in each group). Hamsters were assigned to low-fat diet group (5% *w*/*w* soybean oil, 3.85 kcal/g) and high-fat diet group (35% *w*/*w* soybean oil, 5 kcal/g, 52% of energy) according to the AIN-93M formulation [20] and modification. Therefore, we use 5% soybean oil as the control. One of the groups (C group) was fed the same diet as the sham group (S group) which contained 5% (*w*/*w*) fat from soybean oil. The other three groups were fed with 5%, 15%, and 20% (*w*/*w*) of the experimental oil mixture, namely L, M, and H groups, respectively. The diet consisted of a modified AIN-93 M formula, and the experimental oil mixture consisted of 60% MUFAs and P/S = 5 of a mixture of soybean and canola oils, as shown in Table 2. All animals were fed ad libitum for 8 weeks. Body weight and food intake were measured weekly, and caloric intake and feed efficiency ratio (FER, %; food intake/body weight increment) were calculated. Each group (8 rats) was assigned to three cages, and total food intake of each rat was calculated as the total food amount of three cages divided by eight as average food intake. After 8 weeks of the experiment, the rats were starved for 12 h, and blood was collected under anesthesia with rompun (Bayer, Leverkusen, German) and zoletil (Virbac, Carros, France). Serum was collected centrifuged at 3500× *g* for 20 min at 4 °C. The liver, kidneys, spleen, uterus, and WATs from gonadal, retroperitoneal, and perirenal locations were removed and weighed, and all samples were stored at −80 °C until analyzed.

### 2.3. Serum Measurements

Serum TC, TGs, HDL-C, LDL-C, nonesterified FAs, and glucose levels were analyzed by enzymatic colorimetric analysis using commercial enzyme kits (Randox Laboratory, Crumlin, Northland, UK); serum insulin concentration was analyzed using commercially available enzyme-linked immunosorbent assay (ELISA) kits (Mercodia, Uppsala, Sweden); serum leptin concentration was analyzed using a Mouse and Rat Leptin ELISA Kit (BioVender, Brno, Czech Republic); serum adiponectin concentration was analyzed using a Rat Adiponectin (ACRP30) ELISA Kit (AssayPro, St. Charles, MI, USA); serum estradiol concentration was analyzed using a Rat Estradiol ELISA Kit (Wuhan Fine Biotech, Wuhan, China); and serum follicular-stimulating hormone (FSH) concentration was analyzed using an FSH (Rodent) ELISA Kit (Wuhan Fine Biotech, Wuhan, China) using a VERSAmax microplate reader (Abnova, Taipei, Taiwan). The IR index was estimated by a homoeostasis model assessment (HOMA).

### 2.4. Hepatic Lipid Measurements

Total lipids were extracted from the liver, as previous method [21]. Hepatic cholesterol and TG levels were estimated by commercial enzyme kits (Randox Laboratory); and hepatic nonesterified FA level was estimated by commercial enzyme kits (Randox Laboratory).

### 2.5. Real-Time Euantitative Polymerase Chain Reaction (qPCR)

Total RNA was extracted using Trizol reagent (Life Technologies, Carlsbad, CA, USA) from rat liver and gonadal adipose tissues. The mRNA levels of ACC, ACO, FAS, CPT, AMPK, PPARα, LPL, HSL and PPARγ were quantified by quantitative real-time PCR following Yang’s method [17]. GAPDH mRNA was the internal control. The primers were shown in Table 3.

### 2.6. Hepatic Enzyme Assay and Adipose Tissue Lipoprotein Lipase (LPL) Enzyme Assay

The activities of hepatic lipogenic enzymes (FAS, ACC, ACO, and CPT-I) were measured according to previous protocols [22,23,24,25]. The activity of adipose LPL was measured following Yang’s method [17].

### 2.7. Statistical Analysis

Data are presented as the mean ± standard deviation (SD). Significant differences between the S and C groups were analyzed statistically by Student’s *t*-test to determine changes after an OVX. Significant changes between OVX groups were analyzed by a one-way analysis of variance (ANOVA). When significant changes were observed in ANOVA tests, Tukey’s tests were applied to locate the source of the significant difference. The significance level was set to *p* < 0.05. These calculations were performed using SAS version 9.4 (SAS Institute, Cary, NC, USA).

## 3. Results

### 3.1. Uterine Weight and Hormone-Related Variables

To assess the success of OVX, uterine weight and hormone-related variables were measured. Ten days after the OVX, serum estradiol level was greater in the S group (*n* = 8) than in the OVX group (*n* = 32) (57.52 ± 7.09 vs. 15.06 ± 4.27 ng/mL). After 8 weeks of the dietary intervention, serum estradiol level of the C group was lower than that in the S group (5.56 ± 0.76 vs. 28.88 ± 8.10 ng/mL), with a significantly lower uterine weight (0.10 ± 0.04 vs. 0.56 ± 0.08 g), suggesting the effectiveness of the OVX, which lowered circling estradiol level due to uterine atrophy.

### 3.2. BW, Relative Organ Weights, Food Consumption, and Food Conversion Efficiency

To investigate the effect of dietary fatty acid composition on the basic physiological indexes of rats, we recorded the body weight, relative organ weights and food intake of animals over 8 weeks. At the baseline, there were no significant differences between the S and C groups, or among OVX groups given different dietary regimens. After the recovery period, BWs of the C group were significantly greater than those of the S group, and significance was maintained throughout the experimental period. After 8 weeks of the dietary intervention, the C group had greater BWs than the S group; however, BWs did not differ among the different dietary OVX groups (Table 4). 

Over the 8 weeks, there were no significant differences in the cumulative energy intake over the 8 weeks between the S and C groups, however, M and H groups were significantly greater than that of the C group (Table 5). However, we discovered that the food efficiency of the C group was greater than that of the S group, while it did not differ among the different dietary OVX groups (Table 4).

Spleen, gonad fat, retroperitoneal fat and perirenal fat relative weights did not differ between the S and C groups. However, liver and kidney relative weights of C group were lower than S group. There is no different in liver, kidney, spleen, retroperitoneal fat and perirenal fat relative weights among the OVX groups given different dietary regimens. Gonad fat relative weight of the M group was significantly increased than that of the C group (Table 6).

### 3.3. Serum Lipid Variables and Glucose

To study the effects of dietary fatty acid composition on ameliorating lipid metabolism disorder, plasma lipid profiles including TC, TGs, HDL-C, LDL-C and FFAs were measured. 

After 8 weeks of the dietary intervention, serum TC and LDL-C levels were greater in the C group than in the S group. Among OVX rats, groups fed the low and middle dosages of the experimental oil showed slightly lower levels of TGs and FFAs than the C group. However, the high-dosage (H) group had significantly lower serum TG and FFA concentrations than the C group (Table 7).

Fasting glucose, insulin and HOMA-IR were measured to assess insulin resistance. Fasting glucose and insulin concentrations were higher in the C group than in the S group, with the same results for HOMA-IR. Concentrations of fasting glucose did not differ among the OVX groups. The serum insulin concentration was lower in the L group than in the C group, while there was no significant difference among groups fed different dosages of the experimental oil (Table 7). 

For adipokines, serum leptin level was significantly higher in the C group than in the S group, while serum adiponectin did not differ. In comparing among the OVX groups, we discovered that there were no significant differences in serum leptin concentrations, and serum adiponectin concentrations of the H group were significantly lower than those of the L group. Although there were no significant differences among groups fed different dosages of the experimental oil, adiponectin concentrations of the L, M, and H groups exhibited a decreasing trend with an increase in the dosage of the experimental oil (Table 7).

### 3.4. Hepatic Lipid Metabolism-Related mRNA Expressions

To investigate the effect of dietary fatty acid composition on hepatic lipid metabolism were shown in Figure 1. For the FA synthesis pathway, FAS and ACC mRNA expressions of the S and C groups did not differ, although there was a trend of the OVX increasing their expressions. When comparing different dietary OVX groups, given the same proportion of fat in the diet, FAS mRNA expressions in the L and C groups did not differ. However, the L group had lower ACC mRNA expression than the C group, while given the high dosage of experimental oil in the HFD group, mRNA expressions of FAS and ACC of the H group did not differ from those of the C group (Figure 1A,B).

For the FA oxidation pathway, ACO mRNA expressions did not differ between the S and C groups, while CPT-1 mRNA expressions of the C group were significantly lower than those of the S group. Low and middle dosages of the experimental oil did not make a difference in ACO mRNA expressions, but the high dosage of the experimental oil with an HFD increased ACO mRNA expression. Meanwhile, rats fed the middle and high dosages of the experimental oil had significantly higher CPT-1 mRNA expression, which suggested the experimental oil with 60% MUFAs and P/S = 5 upregulated FA oxidation and altered lipid metabolism in OVX rats (Figure 1C,D).

The mRNA expressions of AMPK were significantly higher in the C group than in the S group; while mRNA expressions of PPARα were significantly lower in the C group than in the S group. After consuming the experimental oil, the L, M, and H groups had significantly lower AMPK and higher PPARα mRNA expressions than the C group (Figure 1E,F).

### 3.5. Hepatic Lipid and Enzyme Activities

To determine whether dietary fatty acid could attenuate the hepatic lipid (liver TC, TG, and FFA concentrations) and enzyme activities (FAS, ACC, ACO, and CPT-1) were examined, as shown in Table 8 and Figure 2. Liver TC, TG, and FFA concentrations, and FAS, ACC, ACO, and CPT-1 activities did not differ between the S and C groups (Table 8, Figure 2). Among the different dietary OVX groups, liver TC and FFA concentrations did not differ. Liver TG concentrations of the M and H groups were significantly lower than those of the C group (Table 8). ACC activity was lower in the L group than in the C group; in contrast, ACO activity was higher in the L group than in the C group. When comparing the C group and the high dosage of experimental oil, the H group, had significantly lower FAS activity, and significantly higher ACO and CPT-1 activities, which suggested suppression of the FA synthesis pathway and promotion of FA oxidation. Meanwhile, we also discovered that although not reaching significance, FAS activity trended lower and CPT-1 activity trended higher with increase dosages of the 60% MUFA and P/S = 5 experimental oil (Figure 2).

### 3.6. Adipose Tissue Enzyme Activities and mRNA Expressions

To investigate the effect of dietary fatty acid composition on adipose tissue enzyme activity (LPL activity) and mRNA expressions (LPL, HSL and PPAR-γ) were shown in Figure 3. Gonad fat LPL activity did not differ between the S and C groups, or among OVX groups given different dietary regimens (Figure 3A). LPL mRNA expressions of the C group were significantly greater than those of the S group, while there were no significant differences among the C, L, M, and H groups. HSL mRNA expressions did not differ between the S and C groups. The M and H groups had greater HSL mRNA expressions than the C and L groups. PPAR-γ mRNA expressions of the C group were significantly lower than those of the S group (Figure 3B–D). However, L, M, and H groups could not lead to recovery of the OVX-induced decrease in PPAR-γ mRNA expression.

## 4. Discussion

In this study, OVX rats in the OVX control group (C group) had significantly lower estrogen level and lower uterine weight, accompanied by uterine atrophy, compared to the sham-operated (S) group, indicating that the operation successfully simulated low estrogen level in menopausal women. Past studies determined that ovarian-related hormones play important roles in regulating energy, appetite, and BW. The low estrogen status of postmenopausal women can easily lead to increased risks of obesity and metabolic-related abnormalities [26]. Similar to results of human experiments, Iwasa et al. [27] found that food intake and BW significantly increased after short-term suppression of ovarian hormone secretion in rats. However, the long-term effects of an OVX on energy metabolism in rats are still inconclusive [28]. In our study, although there were no significant differences in food or energy intake between the OVX and sham (S) group rats, OVX rats still showed a significant increase in BW. This result is similar to experimental results of Jones et al. [29]. Previous results showed that mice with low estrogen level caused by aromatase knockout had significantly heavier BWs and fat weight than wild-type mice, but no significant differences in food intake were seen. In another study, after OVX mice were fed an HFD, it was found that compared to sham-operated mice given the same HFD, their BWs increased by about 27.2% [30], indicating that changes in appetite and food intake were not the necessary factors for body weight gain, as increased food utilization and changes in physical activities may also participate in energy metabolism in OVX animals, which in turn affects body weight changes [31].

Compared to non-menopausal women, postmenopausal women are considered to be more prone to a high body fat percentage and abdominal fat accumulation [8]. In results of past longitudinal studies, postmenopausal women had a 2.88-fold higher risk of developing abdominal obesity than non-menopausal women [32]. Different sex hormones in the body affect the location of fat accumulation in the body. Estrogen in the ovaries is believed to increase the accumulation of peripheral fat in the buttocks and legs, while male hormones are likely to increase accumulation of fat in the abdomen. The decrease of estrogen in the body after menopause causes a relatively high concentration of male hormones in a woman’s body, causing changes in the location of body fat accumulation, and ultimately leading to abdominal fat accumulation [33]. In results of this study, it was found that OVX rats had significant body fat accumulation, which is consistent with excessive fat accumulation in the abdomen under the OVX animal model in past experiments [34]. Past studies found that even under different dietary conditions, OVX animals had increased total fat and abdominal fat [35], reflecting the status of body fat accumulation in postmenopausal women, and the reduction of estrogen in the body after an OVX significantly increased the BW and occurrence of body fat accumulation [36].

It is generally believed that under similar food intake level, HFDs have a higher calorie density per unit of food, which easily causes excessive caloric intake, which then accumulates in the body in the form of TGs, resulting in body weight gain and obesity. However, in this experiment, there were no significant differences in caloric intake among OVX rats given different FA ratios, and there was no higher caloric intake due to the high calorie density of the HFD. After 8 weeks of the intervention with high-dose (20% *w*/*w*) experimental oil (H group) given to OVX rats, BWs did not appear to differ with OVX rats fed a low-dose (5% *w*/*w*) of the experimental oil (L group), indicating that even if diets with different fat-to-total calorie ratios were given, on the basis of isocaloric intake, the 60% MUFA experimental oil with a P/S ratio of 5 did not produce a significant increase in BW. When observing the relative fat weight between groups of OVX rats given different diets, it was found that the relative gonad fat weight of M increased body fat than L and H groups. This phenomenon is consistent with previous studies. In a previous study by Díaz-Rúa et al., 8-week-old Wistar male rats were given a control group diet with fat making up 10% of total calories and an HFD with fat making up 60% of total calories. The daily food intake was controlled so that the two groups reached the same calorie intake. After 4 months of the experiment, the isocaloric HFD group did not a significant increase in body weight, while a higher body fat ratio and a state of obesity, along with increases in fasting plasma glucose and insulin concentrations were seen [37]. According to previous studies, Yang et al. [17] fed DIO hamsters 5%, 15%, and 20% (*w*/*w*) experimental oils for 8 weeks, and found that regardless of the fat percentages in the diet, the experimental oil mixture with 60% MUFAs and P/S = 5 had the effect of preventing body fat accumulation and balancing blood lipids in obese hamsters. Though HFD tends to increase greater BW, due to the results of previous studies, we may believe that regardless of the high percentage of fat content, experimental oil with designed percentage of MUFA may improves the body fat gaining process, which implied that experimental oil may interfere with the fatty acid metabolism pathways.

Under the same food and energy intake conditions, the C group had significantly greater levels of serum TC, LDL-C, fasting blood glucose, insulin, HOMA-IR, and leptin than the S group. There were no significant differences in lipid concentrations or enzyme activities in the liver between the S and the C groups, while lipid oxidation-related mRNA expressions of CPT-1 and PPARα significantly decreased, and AMPK significantly increased. PPAR-α is related to mitochondrial β-oxidation of fatty acids and CPT-1 is a key enzyme that enables fatty acids to go through the inner mitochondrial membrane and reach the mitochondrial matrix to be metabolized [38]. LPL mRNA expression of WATs increased, and mRNA expressions of PPARγ significantly decreased. In summary, an OVX significantly reduced estrogen level and increased insulin concentrations in rats. High concentrations of insulin tend to inhibit the phosphorylation of AMPK, which indirectly inhibits downstream CPT-1 and reduces lipid oxidation. At the same time, it also increases the expression of LPL mRNA and increases adipogenesis in WATs, and ultimately alters lipid metabolism pathways, causing abnormal lipid metabolism in rats.

Under the same percentage of fat sources (5% *w*/*w*), the L group was given experimental oil with 60% MUFAs and P/S = 5. It exhibited significantly reduced serum insulin and HOMA-IR, ACC activity, and mRNA expression in the liver, and increased ACO activity and PPARα mRNA expression in the liver and PPARγ mRNA expression in WATs, compared to the C group. In a previous study, after 4 weeks of diet-induced obesity (DIO), low-fat diets with the same percentage of fat calories (9.5% kcal) were given, and experimental oil consisting of 60% MUFAs and P/S = 5 was found to be more effective in reducing the body fat ratio and plasma TC and leptin concentrations, increasing ACO and CPT-1 activities and mRNA expressions and lipid oxidation in the liver, and reducing body fat accumulation in DIO hamsters compared to those fed soybean oil [18]. The experimental oil with 60% MUFAs and P/S = 5 was confirmed to have the ability to change lipid metabolism pathways in OVX rats by affecting blood glucose-related values, reducing insulin concentrations and lipid synthesis, and increasing lipid oxidation. Without affecting the calorie intake, increasing the lipid-to-calorie ratio in the diet did not increase insulin level or HOMA-IR in the H group of OVX rats fed the HFD. We speculated that the experimental oil mixture of 60% MUFAs and P/S = 5, even under circumstances of an HFD, did not increase the release of FFAs due to excessive intake of fat, nor accumulate in the liver in the form of TGs, resulting in hyperinsulinemia or the occurrence of IR, which is consistent with past studies [16].

In a past study, Yang et al. [17] fed DIO hamsters 5%, 15%, and 20% (*w*/*w*) experimental oils for 8 weeks, and found that regardless of the fat percentages in the diet, the experimental oil mixture with 60% MUFAs and P/S = 5 had the effect of preventing body fat accumulation and balancing blood lipids in obese hamsters. In the results of this experiment, it was also found that BW of rats in the H group that were given an HFD with the experimental oil did not significantly differ from those in the C group, and TG level in the liver and serum TG and FFA concentrations were significantly lower. It was speculated that administration of this HFD, consisting of experimental oil with 60% MUFAs and P/S = 5 did not cause an increase in hepatic insulin level of H group rats. Thus, there was no inhibitory effect of high insulin on AMPK phosphorylation, thereby reducing the inhibitory effect on ACC phosphorylation, and preserving ACC in an inactive state, which would reduce the carboxylation of acetyl CoA and also the production of downstream malonyl-CoA. As a result, the inhibitory effects of malonyl-CoA on CPT-1 were reduced, and the expression of ACO mRNA in the liver increased, consequently reducing the synthesis of hepatic TGs and attenuating hepatic FFA release caused by excessive TG accumulation in the liver [20]. Under an HFD consisting of experimental oil with 60% MUFAs and P/S = 5, it was found that expressions of hepatic PPARα mRNA and PPARγ along with HSL mRNA in WATs significantly increased in the H group, suggesting a trend of the hepatic lipid oxidation pathway, and an increase in lipolysis in adipose tissues, thereby reducing the accumulation of TGs in adipose tissues of OVX rats fed an HFD.

## 5. Conclusions

Nine-week-old female SD rats underwent an OVX, and the decrease in serum estradiol level led to lipid metabolism abnormalities. According to our results, OVX rats had significant weight gain, and also increases in serum total cholesterol, low-density lipoprotein cholesterol, leptin, fasting glucose, insulin, HOMA-IR, and white adipose tissue lipoprotein lipase mRNA expression. After being fed the experimental oil mixture of 60% MUFAs and P/S = 5, decreases in serum triglyceride, free fatty acid, and hepatic triglyceride concentrations were found. Moreover, the H group fed the 20% (*w*/*w*) HFD with the experimental oil mixture exhibited lower fatty acid synthase activity, and higher acetyl CoA oxidase and carnitine-palmitoyltransferase-1 activities and mRNA expressions, along with an increase of white adipose tissue hormone-sensitive lipase mRNA expression. We then concluded that the experimental oil mixture of 60% MUFAs and P/S = 5 may improve abnormalities caused by an OVX through increasing fatty acid oxidation pathways and decreasing fatty acid synthesis pathways.

## Figures and Tables

**Figure 1 nutrients-13-02022-f001:**
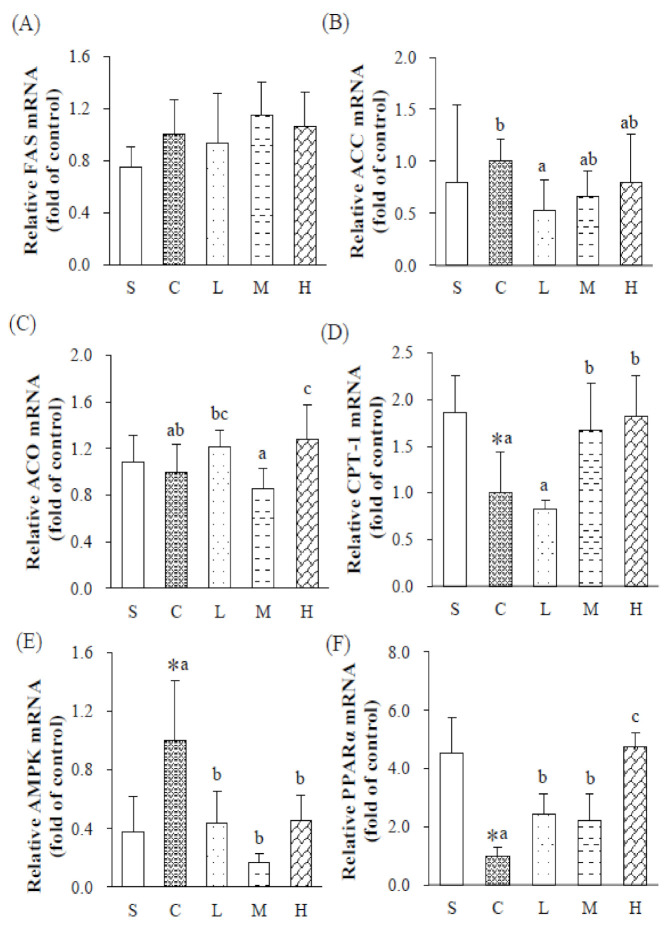
Hepatic mRNA expressions after 8 weeks of a dietary intervention. Relative mRNA expressions of (**A**) acetyl CoA carboxylase (ACC), (**B**) acetyl CoA oxidase (ACO), (**C**) fatty acid synthase (FAS), (**D**) carnitine palmitoyl transferase (CPT)-1, (**E**) 5′ AMP-activated protein kinase (AMPK) and (**F**) peroxisome proliferator-activated receptor alpha (PPARα). Results are presented as the mean ± standard deviation. * *p* < 0.05 vs. S; ^a,b^ *p* < 0.05 between different superscripts. S, sham-operated group; C, control group; L, low-dosage group; M, medium-dosage group; H, high-dosage group.

**Figure 2 nutrients-13-02022-f002:**
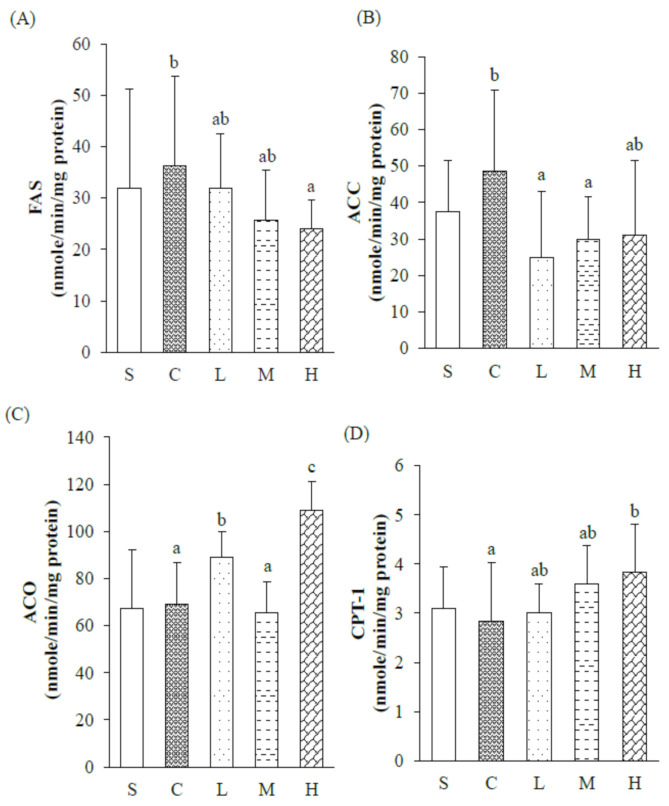
Hepatic lipid metabolic enzyme activities after 8 weeks of a dietary intervention. Hepatic (**A**) fatty acid synthase (FAS) activity, (**B**) acetyl CoA carboxylase (ACC) activity, (**C**) acetyl CoA oxidase (ACO) activity and (**D**) carnitine palmitoyltransferase (CPT)-1 activity. Results are presented as the mean ± standard deviation. ^abc^ *p* < 0.05 between different superscripts. S, sham-operated group; C, control group; L, low-dosage group; M, medium-dosage group; H, high-dosage group.

**Figure 3 nutrients-13-02022-f003:**
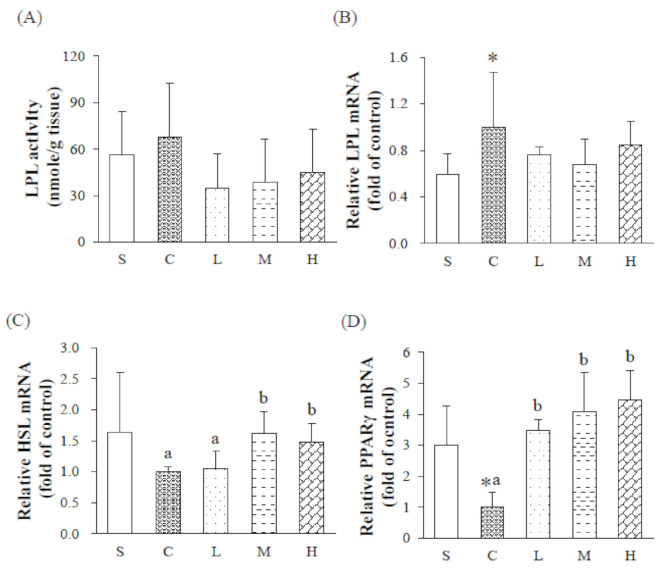
Adipose tissue lipid metabolism variables after 8 weeks of a dietary intervention. (**A**) Adipose tissue lipoprotein lipase (LPL) activity, (**B**) relative LPL mRNA, (**C**) relative hormone-sensitive lipase (HSL) mRNA and (**D**) relative peroxisome proliferator-activated receptor gamma (PPARγ) mRNA. Results are presented as the mean ± standard deviation. * *p* < 0.05 vs. S; ^a,b^ *p* < 0.05 between different superscripts. S, sham-operated group; C, control group; L, low-dosage group; M, medium-dosage group; H, high-dosage group.

**Table 1 nutrients-13-02022-t001:** Fatty acid compositions (in percent, %) of soybean oil and the experimental oil.

	Soybean Oil	Experimental Oil
C14:0	0.49	0.26
C16:0	15.87	5.78
C16:1	0.11	0.43
C17:0	1.24	0.30
C18:0	5.62	1.29
C18:1	19.58	55.18
C18:2	38.67	15.46
C18:3 α	2.90	0.02
C18:3 γ	6.89	8.18
C20:2	5.08	5.17
C20:3	0.22	0
C20:4	1.15	0.16
C20:5	0.18	0.23
C22:4	0.99	0.52
C22:5	0.25	1.15
C22:6	0.77	5.87
Total SFAs	19.97	7.62
Total MUFAs	22.93	55.62
Total PUFAs	57.10	36.76
S/M/P proportion	1: 1.15: 2.85	1: 7.3: 4.82
P/S ratio	2.85	4.82

SFAs, saturated fatty acids; MUFAs, monounsaturated fatty acids; PUFAs, polyunsaturated fatty acids; S/M/P, saturated/monounsaturated/polyunsaturated fatty acids; P/S, polyunsaturated/saturated fatty acids.

**Table 2 nutrients-13-02022-t002:** Composition of the experimental diets (g/kg diet).

Ingredients	Control Diet	L Group	M Group	H Group
Casein	140	140	140	140
L-Cysteine	1.8	1.8	1.8	1.8
Corn starch	610.7	610.7	510.7	460.7
Sucrose	100	100	100	100
Cellulose	50	50	50	50
Soybean oil	50	0	0	0
Experimental oil	0	50	150	200
AIN-93 Mineral Mix	35	35	35	35
AIN-93 Vitamin Mix	10	10	10	10
Choline Bitartrate	2.5	2.5	2.5	2.5
Total weight (g)	1000	1000	1000	1000
Calories (kcal/kg diet)	3563.81	3563.8	4113.8	4388.8
Carbohydrate (% total kcal)	74.9	74.9	56.4	48.8
Fat (% total kcal)	13.7	13.7	33.7	41.9
Protein (% total kcal)	11.4	11.4	9.9	9.3

L, low dosage group; M, medium dosage group; H, high dosage.

**Table 3 nutrients-13-02022-t003:** List of primers used to amplify mRNA by quantitative real-time PCR.

Gene	Accession Number		Sequence (5′→3′)
PPARα	AC_000071.1	Fw	GCTCTGAACATTGGCGTTCG
		Rv	TCAGTCTTGGCTCGCCTCTA
ACC	AC_000078.1	Fw	CTTGGGGTGATGCTCCCATT
		Rv	GCTGGGCTTAAACCCCTCAT
ACO	NC_005101.4	Fw	TGCAGACAGAGACGTAGGAAC
		Rv	AAAGTGGTAGGCACGAATGC
FAS	NC_005109.4	Fw	AGCGGGAAAGTGTACCAGTG
		Rv	GTAGCCGCAGCTCCTTGTAT
CPT1	AC_000069.1	Fw	CACGAAGCCCTCAAACAGATC
		Rv	CCATTCTTGAACCGGATGAAC
AMPK	AC_000070.1	Fw	ATGCCACTTTGCCTTCCGT
		Rv	GCAGTTGCCTACCACCTCAT
PPARγ	NC_005103.4	Fw	GCTGTTATGGGTGAAACTCTGG
		Rv	ATAGGCAGTGCATCAGCGAA
LPL	AC_000084.1	Fw	ATGGCACAGTGGCTGAAAGT
		Rv	CCGGCTTTCACTCGGATCTT
HSL	NC_005100.4	Fw	CTCCTCATGGCTCAACTCC
		Rv	ACTCCTGCGCATAGACTCC
Beta-actin	NC_005111.4	Fw	TGAGCTGCGTTTTACACCCT
		Rv	TTTGGGGGATGTTTGCTCCA

FAS, fatty acid synthase; ACC, acetyl coA carboxylase; ACO, acyl coA oxidase; CPT-1, carnitine palmitoyl transferase-1; AMPK, 5′ AMP-activated protein kinase; PPARα, peroxisome proliferator-activated receptor alpha; LPL, lipoprotein lipase; HSL, hormone sensitive lipase; PPARγ, peroxisome proliferator-activated receptor gamma; Fw, forward; Rv, reverse.

**Table 4 nutrients-13-02022-t004:** Body weight (BW), food intake variables of SD rats after an ovariectomy (OVX) and 8 weeks of a dietary intervention.

Group	S	C	L	M	H
Initial BW (g)	219.6 ± 9.7	217.3 ± 2.5	217.7 ± 5.2	217.5 ± 6.1	217.9 ± 6.7
BW (g)					
4 weeks after OVX	284.9 ± 24.5	337.1 ± 15.6 *	331.2 ± 23.4	331.5 ± 23.0	340.5 ± 20.3
After 8 weeks of dietary intervention	353.5 ± 42.8	416.5 ± 20.5 *	414.9 ± 34.2	450.0 ± 52.4	451.3 ± 48.4
Food intake (g/day/per rat)	19.0 ± 1.1	18.8 ± 3.1	19.6 ± 4.1	18.8 ± 4.1	17.4 ± 3.0
Energy intake (kcal/day/per rat)	67.9 ± 3.7	67.2 ± 10.9	69.9 ± 14.5	77.4 ± 16.9	76.5 ± 15.3
Food efficiency (g BW gain/100 g diet)	6.0 ± 1.7	8.9 ± 2.9 *	9.6 ± 3.4	11.2 ± 4.8	11.3 ± 2.9

Results are presented as the mean ± standard deviation. * *p* < 0.05 vs. S. S, sham-operated group; C, control group; L, low-dosage group; M, medium-dosage group; H, high-dosage group; BW, body weight; OVX, ovariectomy. The food efficiency ratio (FER) was calculated as the total weight gain / total food intake × 100.

**Table 5 nutrients-13-02022-t005:** Energy intake over the course of SD rats after an ovariectomy (OVX) and a dietary intervention.

	Week 1	Week 2	Week 3	Week 4	Week 5	Week 6	Week 7	Week 8	Sum
S	74.9 ± 1.0	63.3 ± 4.4	69.4 ± 1.2	65.0 ± 14.3	73.4 ± 1.9	71.4 ± 1.0	70.2 ± 5.8	65.6 ± 6.6	554.0 ± 4.3 ^a^
C	82.7 ± 9.4	79.8 ± 12.6	76.1 ± 1.3	66.6 ± 0.5	59.6 ± 5.1	58.7 ± 4.0	58.4 ± 5.0	54.1 ± 3.5	537.5 ± 21.8 ^a^
L	86.3 ± 14.5	88.6 ± 5.3	84.3 ± 3.5	68.9 ± 0.3	64.4 ± 7.5	56.8 ± 2.8	55.1 ± 3.0	54.0 ± 1.5	559.1 ± 8.6 ^a^
M	101.4 ± 19.3	101.1 ± 8.7	85.0 ± 0.0	76.5 ± 7.7	70.0 ± 8.8	62.4 ± 0.9	62.5 ± 6.4	60.0 ± 3.5	618.9 ± 2.8 ^b^
H	101.1 ± 10.5	96.1 ± 9.1	81.5 ± 3.6	76.6 ± 8.8	66.9 ± 1.6	65.3 ± 1.7	63.2 ± 0.7	60.2 ± 2.0	611.8 ± 8.3 ^b^

Results are presented as the mean ± standard deviation. ^a,b^ *p* < 0.05 between different superscripts. S, sham-operated group; C, control group; L, low-dosage group; M, medium-dosage group; H, high-dosage group. Sum means the cumulative energy intake.

**Table 6 nutrients-13-02022-t006:** The relative organ weights after an ovariectomy and 8 weeks of a dietary intervention.

	S	C	L	M	H
Liver (%)	2.85 ± 0.59	2.08 ± 0.24 ^*,a^	2.34 ± 0.34 ^a^	2.19 ± 0.28 ^a^	2.33 ± 0.73 ^a^
Kidney (%)	0.62 ± 0.13	0.48 ± 0.06 ^*,a^	0.49 ± 0.04 ^a^	0.47 ± 0.06 ^a^	0.48 ± 0.06 ^a^
Spleen (%)	0.20 ± 0.06	0.19 ± 0.03 ^a^	0.18 ± 0.03 ^a^	0.17 ± 0.01 ^a^	0.16 ± 0.03 ^a^
Gonad fat (%)	2.02 ± 0.66	2.01 ± 0.39 ^a^	2.31 ± 0.54 ^ab^	3.00 ± 0.73 ^b^	2.43 ± 0.57 ^ab^
Retroperitoneal fat (%)	1.97 ± 0.79	2.31 ± 0.35 ^a^	2.29 ± 0.69 ^a^	2.73 ± 0.70 ^a^	2.11 ± 0.38 ^a^
Perirenal fat (%)	1.16 ± 0.38	1.12 ± 0.39 ^a^	1.07 ± 0.26 ^a^	1.26 ± 0.40 ^a^	1.07 ± 0.25 ^a^

Results are presented as the mean ± standard deviation, *n* = 8. * *p* < 0.05 vs. S; ^a,b^ *p* < 0.05 between different superscripts. S, sham-operated group; C, control group; L, low-dosage group; M, medium-dosage group; H, high-dosage group. The relative organ weight = g/g body weight.

**Table 7 nutrients-13-02022-t007:** Serum lipid variables, glucose and TNF-α after an ovariectomy and 8 weeks of a dietary intervention.

	S	C	L	M	H
TC (mg/dL)	85.27 ± 15.94	146.84 ± 25.66 *	117.10 ± 23.73	120.90 ± 23.45	107.33 ± 24.67
TGs (mg/dL)	105.98 ± 24.00	113.78 ± 12.47 ^b^	83.09 ± 4.63 ^ab^	100.12 ± 20.05 ^b^	81.32 ± 14.42 ^a^
HDL-C (mg/dL)	53.68 ± 16.84	43.44 ± 3.35	62.08 ± 18.85	56.32 ± 8.48	48.88 ± 10.62
LDL-C (mg/dL)	12.01 ± 9.50	69.16 ± 21.85 *	40.87 ± 12.03	46.28 ± 35.28	40.04 ± 25.89
FFAs (mmol/dL)	7.22 ± 2.58	10.08 ± 5.02 ^b^	9.30 ± 4.84 ^b^	6.55 ± 4.00 ^ab^	4.80 ± 1.77 ^a^
Fasting glucose (mg/dL)	109.98 ± 7.91	129.83 ± 18.30 *	121.42 ± 11.15	131.60 ± 22.49	132.13 ± 20.05
Insulin (μg/L)	0.15 ± 0.10	0.28 ± 0.07 *^b^	0.18 ± 0.10 ^a^	0.25 ± 0.10 ^ab^	0.22 ± 0.09 ^ab^
HOMA-IR	0.97 ± 0.66	2.29 ± 0.76 *^b^	1.41 ± 0.84 ^a^	1.99 ± 0.73 ^ab^	1.85 ± 0.90 ^ab^
Adiponectin (µg/mL)	6.98 ± 2.79	7.93 ± 1.33 ^ab^	10.31 ± 3.32 ^b^	8.38 ± 2.70 ^ab^	7.70 ± 1.81 ^a^
Leptin (mg/mL)	4.36 ± 2.90	12.03 ± 3.40 *	9.83 ± 4.24	14.80 ± 9.62	11.31 ± 7.36
TNF-α (pg/mL)	2.94 ± 1.45	2.48 ± 0.70	3.36 ± 2.37	2.59 ± 0.73	3.13 ± 0.74

The results represent as mean ± standard deviation (SD), *n* = 8. * *p* < 0.05 vs S; ^a, b^
*p* < 0.05 between different superscripts. TC, total cholesterol; TGs, triglycerides; HDL-C, high-density lipoprotein cholesterol; LDL-C, low-density lipoprotein cholesterol; FFAs, free fatty acids; HOMA-IR, homeostatic model assessment of insulin resistance; TNF, tumor necrosis factor.

**Table 8 nutrients-13-02022-t008:** Hepatic lipid variables after an ovariectomy and 8 weeks of a dietary intervention.

	S	C	L	M	H
Liver TGs (mg/g liver)	11.63 ± 0.67	11.88 ± 0.37 ^b^	11.80 ± 1.14 ^b^	10.74 ± 0.62 ^a^	11.06 ± 0.81 ^a^
Liver TC (mg/g liver)	14.09 ± 0.61	14.27 ± 0.45	13.98 ± 0.26	14.01 ± 0.34	13.99 ± 0.32
Liver FFAs (mmol/g liver)	0.26 ± 0.14	0.27 ± 0.19	0.35 ± 0.09	0.28 ± 0.17	0.26 ± 0.12

The results represent as mean ± standard deviation (SD), *n* = 8. ^a, b^ *p* < 0.05 between different superscripts. S, sham-operated group; C, control group; L, low-dosage group; M, medium-dosage group; H, high-dosage group. TGs, triglycerides; TC, total cholesterol; FFAs, free fatty acids.

## Data Availability

The data that support the findings of this study are available from the corresponding author upon reasonable request.

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
