# Peer review of "Effects of Dietary Fatty Acid Composition on Lipid Metabolism and Body Fat Accumulation in Ovariectomized Rats"

_nutrients, 2021, doi:10.3390/nu13062022_

Round 1
Reviewer 1 Report
Effects of Dietary Fatty Acid Composition on Lipid Metabolism and Body Fat Accumulation in Ovariectomized Rats
Summary:
Yeh and colleagues set out to investigate if diets with low, medium, and high levels of an experimental oil containing a high level of MUFAs would impact lipid metabolism in ovariectomized mice. The authors demonstrate that OVX mice, compared to sham, exhibit higher body weight, fat mass, insulin, and leptin levels. Interestingly, high MUFA levels tend to affect lipogenic and fatty acid oxidation enzymatic activity and overall lipids. Overall, while the data are interesting, there are several concerns that should be addressed.
Major Concerns:
- If 3 rats were assigned to a single cage, how was food intake recorded?
- Similarly, more information on how energy intake and food efficiency were recorded in the materials/methods is needed.
- What is “food efficiency” referring to in Table 3?
- Table 6 is absent
- The results sections do not contain any reasoning as to why certain experiments were conducted, thus it makes it difficult to follow. I would suggest prefacing each section with specific reasons as to why the following experiments were done. For example, in section 3.6, the authors should consider telling the readers why LPL is important to measure and what this could tell us in the grand scheme of the story.
- Activity levels of these enzymes are a nice addition to the story. However, there are several pieces of data that don’t tell us too much. For example, AMPK, PPARα, and HSL are highly regulated at the protein level so steady-state RNA levels aren’t too informative. If these data are included, they should be measured at the protein level.
- Given there are significant differences in body weight after OVX, organ weights should also be presented as a percentage (g/g body weight*100).
- On line 395 it was stated that there were no significant differences in caloric intake among OVX rats given different FA ratios – how is this accurate? According to Table 3, OVX mice consume similar grams of diet per day; yet, the energy content is much greater (4388 kcal/kg vs 3500 kcal/kg) so what is explaining this discrepancy? Energy intake over the course of the study should be calculated and presented; a 10 kcal difference a day is 12% of their daily calories which could have a considerable impact over the course of the study.
Minor Concerns:
- Why do the authors believe the high experimental oil group have less fat weight than the medium oil group?
- Grammatical error on line 274
- Better to present the adiponectin data as µg/mL to minimize the large numbers in Table 5.
- References need to be included with some of the statements made in the discussion (line 420-422, for example).
Author Response
Thanks for your excellent review and comments. We have incorporated the necessary changes in the revised manuscript point by point based on your comments. We have highlighted the changes in the original manuscript by using the track changes mode and the red-colour text.
Q1. If 3 rats were assigned to a single cage, how was food intake recorded?
Ans: We have added them in the revised manuscript (P4, Lines 155-157) as following:
“Each group (8 rats) was assigned to 3 cages, and total food intake of each rat was calculated as the total food amount of 3 cages divided by eight as average food intake.”.
Q2. Similarly, more information on how energy intake and food efficiency were recorded in the materials/methods is needed.
Ans: We have added them in the materials/methods of the revised manuscript (P4, Lines 154-155) as following:
“Body weight and food intake were measured weekly, and caloric intake and feed efficiency ratio (FER, %; food intake/body weight increment) were calculated.”
Q3. What is “food efficiency” referring to in Table 3?
Ans: We have added it in the figure legend of Table 3 of the revised manuscript (P6, Table 3) as following:
The food efficiency ratio (FER) was calculated as the total weight gain/total food intake × 100.
Q4. Table 6 is absent.
Ans: Sorry for that, we have added the table in the revised manuscript (P10)
Q5. The results sections do not contain any reasoning as to why certain experiments were conducted, thus it makes it difficult to follow. I would suggest prefacing each section with specific reasons as to why the following experiments were done. For example, in section 3.6, the authors should consider telling the readers why LPL is important to measure and what this could tell us in the grand scheme of the story.
Ans: We have added them in each section of results in the revised manuscript (P5, Lines 202-203; P6, Lines 210-212; P7, Lines 242-244; P7, Line 249; P8, Lines 270-271; P9, Lines 300-302; P10, Lines 327-329)
Q6. Activity levels of these enzymes are a nice addition to the story. However, there are several pieces of data that don’t tell us too much. For example, AMPK, PPARα, and HSL are highly regulated at the protein level so steady-state RNA levels aren’t too informative. If these data are included, they should be measured at the protein level.
Ans: AMPK involves fatty acid synthesis and fatty acid oxidation. In the study, we evaluate the gene expressions of the liver involved in lipogenesis (FAS and ACC) and lipid oxidation (ACO, p-AMPK, PPAR-α, and CPT1). However, we only selected two cytokines (FAS and ACC) involved in lipogenesis and two cytokines (ACO and CPT1) involved in lipid oxidation. In gene expressions or cytokines of the liver, we found the experimental oil mixture with 60% MUFAs and P/S=5 may reduce lipogenesis and increase lipid oxidation. In this study, the protein of gonadal fat tissue was not easy to extract. Therefore, we did not evaluate the protein expression of HSL and PPARγ.
Q7. Given there are significant differences in body weight after OVX, organ weights should also be presented as a percentage (g/g body weight*100).
Ans: We have modified the table of revised manuscript (P7).
Q8. On line 395 it was stated that there were no significant differences in caloric intake among OVX rats given different FA ratios – how is this accurate? According to Table 3, OVX mice consume similar grams of diet per day; yet, the energy content is much greater (4388 kcal/kg vs 3500 kcal/kg) so what is explaining this discrepancy? Energy intake over the course of the study should be calculated and presented; a 10 kcal difference a day is 12% of their daily calories which could have a considerable impact over the course of the study.
Ans:
- Calorie intake was calculated by the diet intake of each group, though with the similar amount food intake of between different dosage groups, after statistical analysis the average calorie intake between different energy content did not reach significant difference. Although there wasn’t significant difference in energy among groups, we could observe that there was a mild greater BW gain in the M and H group after 8 weeks of dietary intervention.
|
Group |
S |
C |
L |
M |
H |
|
Calories (kcal/kg diet) |
3563.81 |
3563.81 |
3563.8 |
4113.8 |
4388.8 |
|
Food intake (g/day/per rat) |
19.0 ± 1.1 |
18.8±3.1 |
19.6±4.1 |
18.8±4.1 |
17.4±3.0 |
|
Energy intake (kcal/day/per rat) |
67.9 ± 3.7 |
67.2±10.9 |
69.9±14.5 |
77.4±16.9 |
76.5±15.3 |
- We have added energy intake over the course of the study as Table 4 in the revised manuscript (P6).
Table 4. Energy intake over the course of SD rats after an ovariectomy (OVX) and a dietary intervention.
|
|
Week 1 |
Week 2 |
Week 3 |
Week 4 |
Week 5 |
Week 6 |
Week 7 |
Week 8 |
|
S |
74.9±1.0 |
63.3±4.4 |
69.4±1.2 |
65.0±14.3 |
73.4±1.9 |
71.4±1.0 |
70.2±5.8 |
65.6±6.6 |
|
C |
82.7±9.4 |
79.8±12.6 |
76.1±1.3 |
66.6±0.5 |
59.6±5.1 |
58.7±4.0 |
58.4±5.0 |
54.1±3.5 |
|
L |
86.3±14.5 |
88.6±5.3 |
84.3±3.5 |
68.9±0.3 |
64.4±7.5 |
56.8±2.8 |
55.1±3.0 |
54.0±1.5 |
|
M |
101.4±19.3 |
101.1±8.7 |
85.0±0.0 |
76.5±7.7 |
70.0±8.8 |
62.4±0.9 |
62.5±6.4 |
60.0±3.5 |
|
H |
101.1±10.5 |
96.1±9.1 |
81.5±3.6 |
76.6±8.8 |
66.9±1.6 |
65.3±1.7 |
63.2±0.7 |
60.2±2.0 |
Q9. Why do the authors believe the high experimental oil group have less fat weight than the medium oil group?
Ans: We have added them in the discussion of revised manuscript (P12, Lines 401-408) as following.
According to previous studies, Yang et al. (2017) fed DIO hamsters 5%, 15%, and 20% (w/w) experimental oils for 8 weeks, and found that regardless of the fat percentages in the diet, the experimental oil mixture with 60% MUFAs and P/S=5 had the effect of preventing body fat accumulation and balancing blood lipids in obese hamsters. Though HFD tends to increase greater BW and fat weight, due to the results of previous studies, we may believe that regardless of the high percentage of fat content, experimental oil with designed percentage of MUFA may improves the body fat gaining process, which implied in the further part of the manuscript, that experimental oil may interfere with the fatty acid metabolism pathways.
Reference:
Yang, S.C.; Lin, S.H.; Chang, J.S.; Chien, Y.W. High fat diet with a high monounsaturated fatty acid and polyunsaturated/saturated fatty acid ratio suppresses body fat accumulation and weight gain in obese hamsters. Nutrients 2017, 9, p. 1148.
Q10. Grammatical error on line 274
Ans: We have modified them in the revised manuscript (P4, Lines 245-248) as following.
“Among OVX rats, groups fed the low and middle dosages of the experimental oil showed slightly lower levels of TGs and FFAs than the C group. However, the high-dosage (H) group had significantly lower serum TG and FFA concentrations than the C group (Table 6).”
Q11. Better to present the adiponectin data as µg/mL to minimize the large numbers in Table 5.
Ans: We have presented the adiponectin data as µg/mL in Table 5 of the revised manuscript (P8, Table 6)
Q12. References need to be included with some of the statements made in the discussion (line 420-422, for example).
Ans: I have added them in the discussion and reference in the revised manuscript (P12, Lines 401-408; P12, Lines 413-416).
Reference
Yang, S.C.; Lin, S.H.; Chang, J.S.; Chien, Y.W. High fat diet with a high monounsaturated fatty acid and polyunsaturated/saturated fatty acid ratio suppresses body fat accumulation and weight gain in obese hamsters. Nutrients 2017, 9, p. 1148.
Serviddio, G.; Giudetti, A.M.; Bellanti, F.; Priore, P.; Rollo, T.; Tamborra, R.; Siculella, L.; Vendemiale, G.; Altomare, E.; Gnoni, G.V. Oxidation of hepatic carnitine palmitoyl transferase-I (CPT-I) impairs fatty acid beta-oxidation in rats fed a methionine-choline deficient diet. PLoS One. 2011, 6, e24084.
Reviewer 2 Report
Very well planned experiment showing all trends in a satisfactory way. Maybe not in all places statistical significance was found, but it is the reality of surgical interventions in rats. Maybe, in future, choosing 12 weeks rats would be more optimal. It is easier to choose animals with tendency to be bigger/smaller. In general optimal methodology and conclusions.
Author Response
Q1. Maybe not in all places statistical significance was found, but it is the reality of surgical interventions in rats.
Ans: Thanks for your excellent review and comments. In fact, the standard derivation of surgical interventions is larger. Therefore, not in all places statistical significance was found.
Q2. Maybe, in future, choosing 12 weeks rats would be more optimal. It is easier to choose animals with tendency to be bigger/smaller. In general optimal methodology and conclusions.
Ans: Thank you for your comment. Many studies have chosen to perform ovariectomy on 10-week-old mice to induce female osteoporosis (Guo et al., 2012; Mohamed et al., 2000). Therefore, I think the age of the mouse is appropriate.
Reference
Guo, Y.; Li, M.; Zhusheng, L.; Yamada, T.; Sasaki, M.; Hasegawa, T.; Hongo, H.; Tabata, C.; Suzuki, R.; Oda, K.; Yamamoto, T.; Kawanami, M.; Amizuka, N. Immunolocalization of sclerostin synthesized by osteocytes in relation to bone remodeling in the interradicular septa of ovariectomized rats. J. Electron. Microsc. (Tokyo). 2012, 61, 309-20.
Mohamed, M.K.; Abdel-Rahman, A.A. Effect of long-term ovariectomy and estrogen replacement on the expression of estrogen receptor gene in female rats. Eur. J. Endocrinol. 2000, 142, 307-314.
Reviewer 3 Report
This is an interesting and well written manuscript. However, the effects of various fats on body weight and fat metabolism have been well studied including humans and animal models. Therefore, the results provide only limited new information and thus scientific value for publication in this journal. Indeed, some of the results are very conceivable. For example, the body weight and calorie intake of M and H fat groups certainly would be relatively higher than L and C controls. Given this, the better experimental design should be to give all treatment groups with the same amount of fat but only variance in different fat acid composition, which could potentially provide some novelty and clinical significance for preventing obesity in postmenopausal women. Several specific comments are below:
Please include the rationale for the use of 5% soybean oil as the control.
Please given the detail of experimental oil mixture.
Please explain why BW variation is so high in M and H group after 8 weeks of treatment, but not initial or 4 weeks?
Author Response
Thanks for your excellent review and comments. We have incorporated the necessary changes in the revised manuscript point by point based on your comments. We have highlighted the changes in the original manuscript by using the track changes mode and the red-colour text.
Q1. Please include the rationale for the use of 5% soybean oil as the control.
Ans: Hamsters were assigned to low-fat diet group (5% w/w soybean oil, 3.85 kcal/g) and high-fat diet group (35% w/w soybean oil, 5 kcal/g, 52% of energy) according to the AIN-93M formulation (Yang et al., 2016) and modification. Therefore, we use 5% soybean oil as the control. I have modified them in the material and method in the revised manuscript (P4, Lines 145-147).
Reference
Yang, J.H.; Chang, J.S.; Chen, C.L.; Yeh, C.L.; Chien, Y.W. Effects of different amounts and types of dietary fatty acids on the body weight, fat accumulation, and lipid metabolism in hamsters. Nutrition 2016, 32, p. 601–608.
Q2. Please given the detail of experimental oil mixture.
Ans: The diet consisted of a modified AIN-93 M formula, and the experimental oil mixture consisted of 60% MUFAs and P/S=5 of a mixture of soybean and canola oils, as shown in Table 2.
Q3. Please explain why BW variation is so high in M and H group after 8 weeks of treatment, but not initial or 4 weeks?
Ans: At baseline, the rats were grouped according to the S distribution. Therefore, there were no significant differences between all groups and BW variation was small. After a 4-week surgical recovery period, OVX rats were divided into four different dietary groups according to the S distribution. Therefore, there is no significant difference in BW among four different dietary groups and BW variation was small. After 8 weeks of diet intervention, because 1. The intervention time is long, the BW variation is greater; 2. The greater BW of the rat, the greater the relative SD.
Round 2
Reviewer 1 Report
Comment 1: qPCR primer sequences should be provided.
Comment 2: The table 4 that was included is a nice addition; however, it still does not show the cumulative food intake over the course of the whole experiment. It would be good for the readers to know the cumulative energy intake, per rat, over the 8 weeks and to determine if their are significant differences between groups.
Comment 3: The authors note they measured cytokines in the liver in response to my queries in the original version. Just a side comment, these lipogenic and FAO genes that were measured aren't considered cytokines.
Author Response
Thanks for your excellent review and comments. We have incorporated the necessary changes in the revised manuscript point by point based on your comments. We have highlighted the changes in the original manuscript by using the track changes mode and the red-colour text.
Comment 1: qPCR primer sequences should be provided.
Response: We have added the information of qPCR primer sequences in Table 3 in the revised manuscript (Page 5, 6).
Comment 2: The table 4 that was included is a nice addition; however, it still does not show the cumulative food intake over the course of the whole experiment. It would be good for the readers to know the cumulative energy intake, per rat, over the 8 weeks and to determine if their are significant differences between groups.
Response: We have added the information of the cumulative energy intake in Table and text in the revised manuscript (Page 6, Lines 227-230; Table 5).
Comment 3: The authors note they measured cytokines in the liver in response to my queries in the original version. Just a side comment, these lipogenic and FAO genes that were measured aren't considered cytokines.
Response: Thank you for your reminder. Actually, we only selected two lipogenic activity (FAS and ACC) and two lipolytic activities (ACO and CPT1). In gene expressions or activities of the liver, we found the experimental oil mixture with 60% MUFAs and P/S=5 may reduce lipogenesis and increase lipid oxidation.
Reviewer 3 Report
This revise manuscript has been significantly improved in writing and data presenting, but still there is still lack of scientific merit, as the effect of fat and its fatty acid composition on body weight, fat and glucose metabolism has been well studied, and the paper in the current form failed to provide new information beyond what has already been known.
Author Response
Comment: This revise manuscript has been significantly improved in writing and data presenting, but still there is still lack of scientific merit, as the effect of fat and its fatty acid composition on body weight, fat and glucose metabolism has been well studied, and the paper in the current form failed to provide new information beyond what has already been known.
Response: Although a number of studies have been carried out to investigate the effect of fat and its fatty acid composition on body weight, fat and glucose metabolism in normal animals, the effects of fatty acid composition in OVX rats have not been studied. Therefore, the study results could provide menopause how to select fatty acid composition. We have added the information in the introduction of the revised manuscript (Page 3, Lines 108-110, 113-114).